# Experience Replay for Continual Learning

**David Rolnick**
University of Pennsylvania
Philadelphia, PA USA
`drolnick@seas.upenn.edu`

**Arun Ahuja**
DeepMind
London, UK
`arahuja@google.com`

**Jonathan Schwarz**
DeepMind
London, UK
`schwarzjn@google.com`

**Timothy P. Lillicrap**
DeepMind
London, UK
`countzero@google.com`

**Greg Wayne**
DeepMind
London, UK
`gregwayne@google.com`

## Abstract

Interacting with a complex world involves continual learning, in which tasks and data distributions change over time. A continual learning system should demonstrate both plasticity (acquisition of new knowledge) and stability (preservation of old knowledge). Catastrophic forgetting is the failure of stability, in which new experience overwrites previous experience. In the brain, replay of past experience is widely believed to reduce forgetting, yet it has been largely overlooked as a solution to forgetting in deep reinforcement learning. Here, we introduce CLEAR, a replay-based method that greatly reduces catastrophic forgetting in multi-task reinforcement learning. CLEAR leverages off-policy learning and behavioral cloning from replay to enhance stability, as well as on-policy learning to preserve plasticity. We show that CLEAR performs better than state-of-the-art deep learning techniques for mitigating forgetting, despite being significantly less complicated and not requiring any knowledge of the individual tasks being learned.

## 1 Introduction

An intelligent system that interacts with its environment faces the problem of *continual learning*, in which new experience is constantly being acquired, while old experience may still be relevant [25]. An effective continual learning system must optimize for two potentially conflicting goals. First, when a previously encountered scenario is encountered, performance should immediately be good, ideally as good as it was historically. Second, maintenance of old skills should not inhibit rapid acquisition of a new skill. These simultaneous constraints – maintaining the old while still adapting to the new – represent the challenge known as the *stability-plasticity* dilemma [6].

The quintessential failure mode of continual learning is *catastrophic forgetting*, in which new knowledge supplants old knowledge, resulting in high plasticity but little stability. Within biological systems, hippocampal replay has been proposed as a systems-level mechanism to reduce catastrophic forgetting and improve generalization, as in the theory of complementary learning systems [18]. This process complements local consolidation of past experience at the level of individual neurons and synapses, which is also believed to be present in biological neural networks [2, 14].

Within artificial continual learning systems, deep neural networks have become ubiquitous tools and increasingly are applied in situations where catastrophic forgetting becomes relevant. In reinforcement learning (RL) settings, this problem is often circumvented by using massive computational resources to ensure that all tasks are learned simultaneously, instead of sequentially. Namely, in simulation or in self-play RL, fresh data can be generated on demand and trained on within the same minibatch, resulting in a stable data distribution.

However, as RL is increasingly applied to continual learning problems in industry or robotics, situations arise where gathering new experience is expensive or difficult, and thus simultaneous training is infeasible. Instead, an agent must be able to learn from one task at a time, and the sequence in which tasks occur is not under the agent's control. In fact, boundaries between tasks will often be unknown – or tasks will deform continuously and not have definite boundaries at all [12]. Such a paradigm for training eliminates the possibility of simultaneously acting upon and learning from several tasks, and leads to the possibility of catastrophic forgetting.

There has recently been a surge of interest in methods for preventing catastrophic forgetting in RL, inspired in part by neuroscience [11, 13, 27, 29]. Remarkably, however, such work has focused on synaptic consolidation approaches, while possibilities of experience replay for reducing forgetting have largely been ignored, and it has been unclear how replay could be applied in this context within a deep RL framework.

We here demonstrate that replay can be a powerful tool in continual learning. We propose a simple technique, Continual Learning with Experience And Replay (CLEAR), mixing on-policy learning from novel experiences (for plasticity) and off-policy learning from replay experiences (for stability). For additional stability, we introduce behavioral cloning between the current policy and its past self. While it can be memory-intensive to store all past experience, we find that CLEAR is just as effective even when memory is severely constrained. Our approach has the following advantages:

- **Stability and plasticity.** CLEAR performs better than state-of-the-art (Elastic Weight Consolidation, Progress & Compress), almost eliminating catastrophic forgetting.

- **Simplicity.** CLEAR is much simpler than prior methods for reducing forgetting and can also be combined easily with other approaches.

- **No task information.** CLEAR does not rely upon the identity of tasks or the boundaries between them, by contrast with prior art. This means that CLEAR can be applied to a much broader range of situations, where tasks are unknown or are not discrete.

## 2   Related work

The problem of catastrophic forgetting in neural networks has long been recognized [6], and it is known that rehearsing past data can be a satisfactory antidote for some purposes [4, 18]. Consequently, in the supervised setting that is the most common paradigm in machine learning, forgetting has been accorded less attention than in cognitive science or neuroscience, since a fixed dataset can be reordered and replayed as necessary to ensure high performance on all samples.

In recent years, however, there has been renewed interest in overcoming catastrophic forgetting in RL and in supervised learning from streaming data, where it is not possible simply to reorder the data (see e.g. [8, 17, 24, 26]). Current strategies for mitigating catastrophic forgetting have primarily focused on schemes for protecting the parameters inferred in one task while training on another. For example, in Elastic Weight Consolidation (EWC) [13], weights important for past tasks are constrained to change more slowly while learning new tasks. Progressive Networks [27] freezes subnetworks trained on individual tasks, and Progress & Compress [29] uses EWC to consolidate the network after each task has been learned. Kaplanis et al. [11] treat individual synaptic weights as dynamical systems with latent dimensions / states that protect information. Outside of RL, Zenke et al. [32] develop a method similar to EWC that maintains estimates of the importance of weights for past tasks, Li and Hoiem [15] leverage a mixture of task-specific and shared parameters, and Milan et al. [19] develop a rigorous Bayesian approach for estimating unknown task boundaries. Notably all these methods assume that task identities or boundaries are known, with the exception of [19], for which the approach is likely not scalable to highly complex tasks.

Rehearsing old data via experience replay buffers is a common technique in RL, but such methods have largely been driven by the goal of data-efficient learning on single tasks [7, 16, 20]. Research in this vein has included prioritized replay for maximizing the impact of rare experiences [28], learning from human demonstration data seeded into a buffer [9], and methods for approximating replay buffers with generative models [30]. A noteworthy use of experience replay buffers to protect against catastrophic forgetting was demonstrated in Isele and Cosgun [10] on toy tasks, with a focus on how buffers can be made smaller. Previous works [7, 22, 31] have explored mixing on- and off-policy updates in RL, though these were focused on speed and stability in individual tasks and did not examine continual learning.

While it may not be surprising that replay can reduce catastrophic forgetting to some extent, it is remarkable that, as we show, it is powerful enough to outperform state-of-the-art methods. There is a marked difference between reshuffling data in supervised learning and replaying past data in RL. Notably, in RL, past data are typically leveraged best by *off-policy* algorithms since historical actions may come from an out-of-date policy distribution. Reducing this deviation is our motivation for the behavioral cloning component of CLEAR, inspired by work showing the power of policy consolidation [12], self-imitation [23], and knowledge distillation [5].

## 3 The CLEAR Method

CLEAR uses actor-critic training on a mixture of new and replayed experiences. We employ distributed training based on the Importance Weighted Actor-Learner Architecture presented in [3]. Namely, a single learning network is fed experiences (both novel and replay) by a number of acting networks, for which the weights are asynchronously updated to match those of the learner. Training proceeds as in [3] by the *V-Trace* off-policy learning algorithm, which uses truncated importance weights to correct for off-policy distribution shifts. While V-Trace was designed to correct for the lag between the parameters of the acting networks and those of the learning network, we find it also successfully corrects for the distribution shift corresponding to replay experience. Our network architecture and training hyperparameters are chosen to match those in [3] and are not further optimized.

Formally, let $\theta$ denote the network parameters, $\pi_\theta$ the (current) policy of the network over actions $a$, $\mu$ the policy generating the observed experience, and $h_s$ the hidden state of the network at time $s$. Then, the V-Trace target $v_s$ is given by:

$$v_s := V(h_s) + \sum_{t=s}^{s+n-1} \gamma^{t-s} \left( \prod_{i=s}^{t-1} c_i \right) \delta_t V,$$

where $\delta_t V := \rho_t \left( r_t + \gamma V(h_{t+1}) - V(h_t) \right)$ for truncated importance sampling weights $c_i := \min(\bar{c}, \frac{\pi_\theta(a_i|h_i)}{\mu(a_i|h_i)})$, and $\rho_t = \min(\bar{\rho}, \frac{\pi_\theta(a_t|h_t)}{\mu(a_t|h_t)})$ (with $\bar{c}$ and $\bar{\rho}$ constants). The policy gradient loss is:

$$L_{\text{policy-gradient}} := -\rho_s \log \pi_\theta(a_s|h_s) \left( r_s + \gamma v_{s+1} - V_\theta(h_s) \right).$$

The value function update is given by the L2 loss, and we regularize policies using an entropy loss:

$$L_{\text{value}} := \left( V_\theta(h_s) - v_s \right)^2, \quad L_{\text{entropy}} := \sum_a \pi_\theta(a|h_s) \log \pi_\theta(a|h_s).$$

The loss functions $L_{\text{policy-gradient}}$, $L_{\text{value}}$, and $L_{\text{entropy}}$ are applied both for new and replay experiences. In addition, we add $L_{\text{policy-cloning}}$ and $L_{\text{value-cloning}}$ for replay experiences only. In general, our experiments use a 50-50 mixture of novel and replay experiences, though performance does not appear to be very sensitive to this ratio. Further implementation details are given in Appendix A.

In the case of replay experiences, two additional loss terms are added to induce behavioral cloning between the network and its past self, with the goal of preventing network output on replayed tasks from drifting while learning new tasks. We penalize (1) the KL divergence between the historical policy distribution and the present policy distribution, (2) the L2 norm of the difference between the historical and present value functions. Formally, this corresponds to adding the loss functions:

$$L_{\text{policy-cloning}} := \sum_a \mu(a|h_s) \log \frac{\mu(a|h_s)}{\pi_\theta(a|h_s)}, \quad L_{\text{value-cloning}} := ||V_\theta(h_s) - V_{\text{replay}}(h_s)||_2^2.$$

Note that computing $\text{KL}[\mu||\pi_\theta]$ instead of $\text{KL}[\pi_\theta||\mu]$ ensures that $\pi_\theta(a|h_s)$ is nonzero wherever the historical policy is as well.

## 4 Results

### 4.1 Catastrophic forgetting vs. interference

Our first experiment (Figure 1) was designed to distinguish between two distinct concepts that are sometimes conflated, *interference* and *catastrophic forgetting*, and to emphasize the outsized role of the latter as compared to the former. Interference occurs when two or more tasks are incompatible (*destructive interference*) or mutually helpful (*constructive interference*) within the same model. Catastrophic forgetting occurs when a task's performance goes down not because of incompatibility

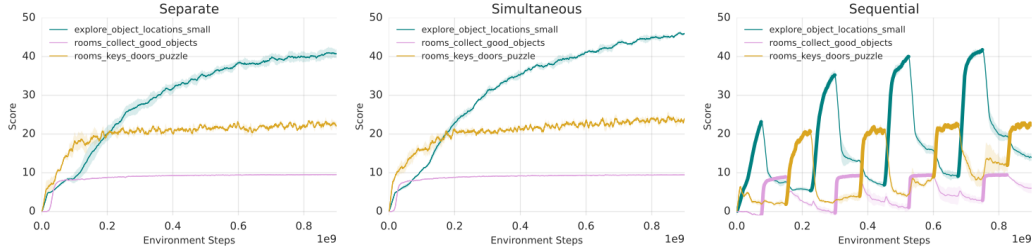

Figure 1: Separate, simultaneous, and sequential training: the $x$-axis denotes environment steps summed across all tasks and the $y$-axis episode score. In "Sequential", thick line segments are used to denote the task currently being trained, while thin segments are plotted by evaluating performance without learning. In simultaneous training, performance on `explore_object_locations_small` is higher than in separate training, an example of modest constructive interference. In sequential training, tasks that are not currently being learned exhibit very dramatic catastrophic forgetting. See Appendix B for plots of the same data, showing cumulative performance.

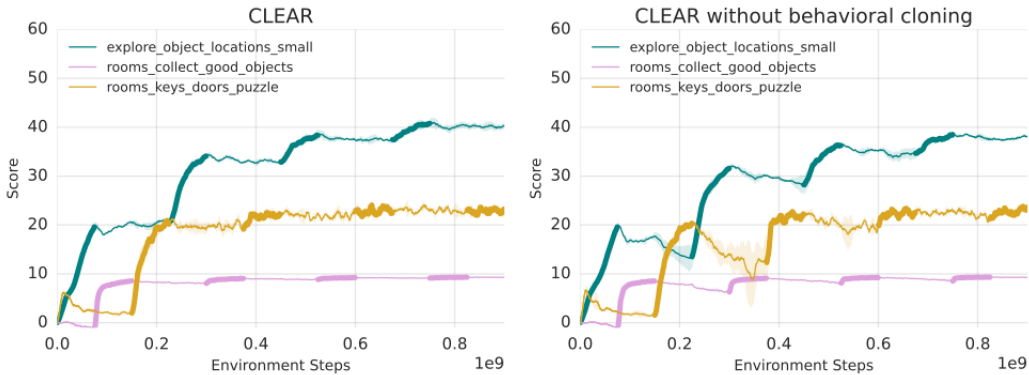

Figure 2: Demonstration of CLEAR on three DMLab tasks, which are trained cyclically in sequence. CLEAR reduces catastrophic forgetting so significantly that sequential tasks train almost as well as simultaneous tasks (compare to Figure 1). When the behavioral cloning loss terms are ablated, there is still reduced forgetting from off-policy replay alone. As above, thicker line segments are used to denote the task that is currently being trained. See Appendix B for plots of the same data, showing cumulative performance.

with another task but because the second task overwrites it within the model. As we aim to illustrate, the two are independent phenomena, and while interference may happen, forgetting is ubiquitous.[1]

We considered a set of three distinct tasks within the DMLab set of environments [1], and compared three training paradigms on which a network may be trained to perform these three tasks: (1) Training networks on the individual tasks *separately*, (2) training a single network examples from all tasks *simultaneously* (which permits interference among tasks), and (3) training a single network *sequentially* on examples from one task, then the next task, and so on cyclically. Across all training protocols, the total amount of experience for each task was held constant. Thus, for separate networks training on separate tasks, the $x$-axis in our plots shows the total number of environment frames summed across all tasks. For example, at three million frames with one million each on tasks one, two, and three. This allows a direct comparison to simultaneous training, in which the same network was trained on all three tasks.

We observe that in DMLab, there is very little difference between separate and simultaneous training. This indicates minimal interference between tasks. If anything, there is slight constructive interference, with simultaneous training performing marginally better than separate training. We assume this is a result of (i) commonalities in image processing required across different tasks, and (ii) certain

basic exploratory behaviors, e.g., moving around, that are advantageous across tasks. (By contrast, destructive interference might result from incompatible behaviors or from insufficient model capacity.)

By contrast, there is a large difference between either of the above modes of training and sequential training, where performance on a task decays immediately when training switches to another task – that is, catastrophic forgetting. Note that the performance of the sequential training appears at some points to be greater than that of separate training. This is purely because in sequential training, training proceeds exclusively on a single task, then exclusively on another task. For example, the first task quickly increases in performance since the network is effectively seeing three times as much data on that task as the networks training on separate or simultaneous tasks.

| | explore... | rooms_collect... | rooms_keys... |
|---|---|---|---|
| Separate | 29.24 | 8.79 | 19.91 |
| Simultaneous | 32.35 | 8.81 | 20.56 |
| Sequential (no CLEAR) | 17.99 | 5.01 | 10.87 |
| CLEAR (50-50 new-replay) | **31.40** | **8.00** | 18.13 |
| CLEAR w/o behavioral cloning | 28.66 | 7.79 | 16.63 |
| CLEAR, 75-25 new-replay | 30.28 | 7.83 | 17.86 |
| CLEAR, 100% replay | 31.09 | 7.48 | 13.39 |
| CLEAR, buffer 5M | 30.33 | **8.00** | 18.07 |
| CLEAR, buffer 50M | 30.82 | 7.99 | **18.21** |

Figure 3: Quantitative comparison of the final cumulative performance between standard training ("Sequential (no CLEAR)") and various versions of CLEAR on a cyclically repeating sequence of DMLab tasks. We also include the results of training on each individual task with a separate network ("Separate") and on all tasks simultaneously ("Simultaneous") instead of sequentially (see Figure 1). As described in Section 4.1, these are no-forgetting scenarios and thus present upper bounds on the performance expected in a continual learning setting, where tasks are presented sequentially. Remarkably, CLEAR achieves performance comparable to "Separate" and "Simultaneous", demonstrating that forgetting is virtually eliminated. See Appendix B for further details and plots.

## 4.2 Stability

We here demonstrate the efficacy of CLEAR for diminishing catastrophic forgetting (Figure 2). We apply CLEAR to the cyclically repeating sequence of DMLab tasks used in the preceding experiment. Our method effectively eliminates forgetting on all three tasks, while preserving overall training performance (see "Sequential" training in Figure 1 for reference). When the task switches, there is little, if any, dropoff in performance when using CLEAR, and the network picks up immediately where it left off once a task returns later in training. Without behavioral cloning, the mixture of new experience and replay still reduces catastrophic forgetting, though the effect is reduced.

In Figure 3, we perform a quantitative comparison of the performance for CLEAR against the performance of standard training on sequential tasks, as well as training on tasks separately and simultaneously. In order to perform a comparison that effectively captures the overall performance during continual learning (including the effect of catastrophic forgetting), the reward shown for time $t$ is the average $(1/t)\sum_{s<t} r_s$, thus effectively measuring the area under the curve for plots such as Figure 2. (We replot the results of our main experiments according to the cumulative performance in Appendix B.) We find that CLEAR attains similar cumulative performance to networks trained on tasks separately and simultaneously – effectively eliminating catastrophic forgetting.

## 4.3 Plasticity

It is a reasonable worry that relying on a replay buffer could cause new tasks to be learned more slowly as the new task data will make up a smaller and smaller portion of the replay buffer as the buffer gets larger. In this experiment (Figure 4), we find that this is not a problem for CLEAR, relying as it does on a mixture of off- and on-policy learning. Specifically, we find the performance attained on a task is largely independent of the amount of data stored in the buffer and on the identities of the preceding tasks. We consider a cyclically repeating sequence of three DMLab tasks. At different points in the sequence, we insert a fourth DMLab task (natlab_varying_map_randomized) as a "probe". We find that the performance attained on the probe task is independent of the point at which it is introduced within the training sequence. This is true both for normal training and for CLEAR. Notably, CLEAR succeeds in greatly reducing catastrophic forgetting for all tasks, and the effect on the probe task does not diminish as the probe task is introduced later on in the training sequence. We also test

the performance of CLEAR using 100% (off-policy) replay experience. We observe that, unlike with standard CLEAR, the performance obtained on the probe task `natlab_varying_map_randomized` deteriorates markedly as it appears later in the sequence of tasks. For later positions in the sequence, the probe task comprises a smaller percentage of replay experience, thereby impeding purely off-policy learning. This result underlines why CLEAR uses new experience, as well as replay, to allow rapid learning of new tasks.

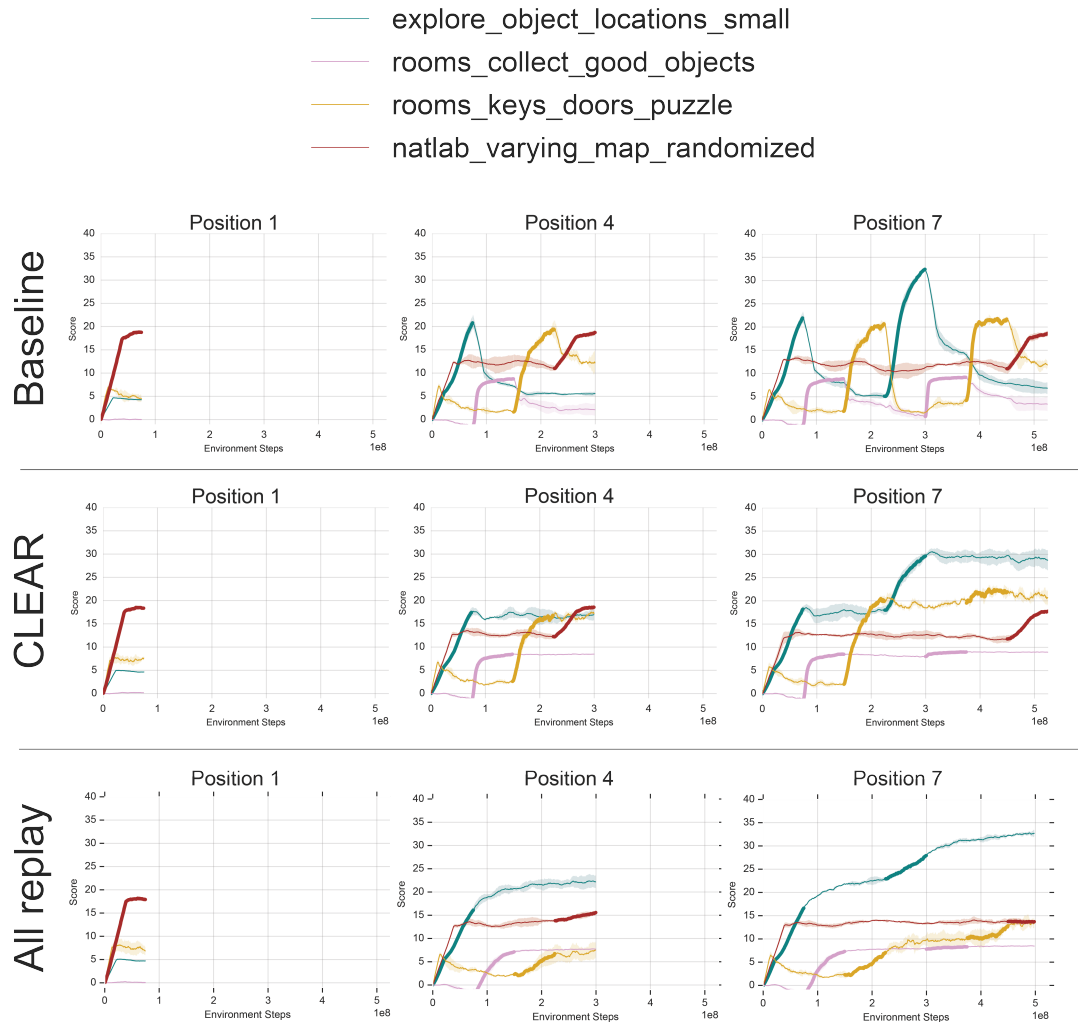

Figure 4: The DMLab task `natlab_varying_map_randomize` (brown line) is presented as a "probe task" at different positions within a cyclically repeating sequence of three other DMLab tasks. As in previous experiments, CLEAR greatly reduces catastrophic forgetting. The key point here is that the performance attained on the probe task does not depend on the identities of the preceding tasks and does not degrade as more experiences and tasks are introduced into the buffer (i.e., as the probe task appears later in the task sequence). In the case that all learning is driven by the replay buffer, however, performance on the probe task *does* degrade as the buffer fills. This justifies our use of a mixture of novel and replay experience in training CLEAR, to allow for fast learning of new tasks, in addition to preventing catastrophic forgetting.

### 4.4 Balance of on- and off-policy learning

In this experiment (Figure 5), we consider the ratio of new examples to replay examples during training. Using 100% new examples is simply standard training, which as we have seen is subject to dramatic catastrophic forgetting. At 75-25 new-replay, there is already significant resistance to forgetting. At the opposite extreme, 100% replay examples is extremely resistant to catastrophic forgetting, but at the expense of a (slight) decrease in performance attained. We believe that 50-50 new-replay represents a good tradeoff, combining significantly reduced catastrophic forgetting

with no appreciable decrease in performance attained. Unless otherwise stated, our experiments on CLEAR will use a 50-50 split of new and replay data in training.

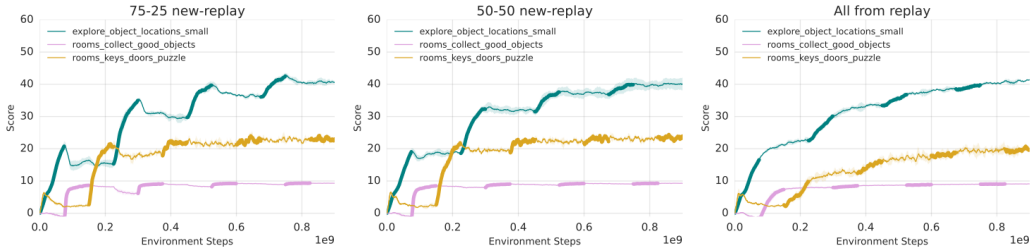

Figure 5: Comparison between different proportions of new and replay examples while cycling training among three tasks. We observe that with a 75-25 new-replay split, CLEAR eliminates most, but not all, catastrophic forgetting. At the opposite extreme, 100% replay prevents forgetting at the expense of reduced overall performance (with an especially noticeable reduction in early performance on the task `rooms_keys_doors_puzzle`). A 50-50 split represents a good tradeoff between these extremes. See Appendix B for plots of the same data, showing cumulative performance.

It is notable that it is possible to train purely on replay examples, since the network has essentially no on-policy learning. In fact, the figure shows that with 100% replay, performance on each task increases throughout, even when on-policy learning is being applied to a different task. Just as Figure 2 shows the importance of behavioral cloning for maintaining past performance on a task, so this experiment shows that off-policy learning can actually increase performance from replay alone. Both ingredients are necessary for the success of CLEAR.

## 4.5 Limited-size buffers

In some cases, it may be impractical to store all past experiences in the replay buffer. We therefore test the efficacy of buffers that have capacity for only a relatively small number of experiences (Figure 6). Once the buffer is full, we use reservoir sampling to decide when to replace elements of the buffer with new experiences [10] (see details in Appendix A). Thus, at each point in time, the buffer contains a (fixed size) sample uniformly at random of all past experiences.

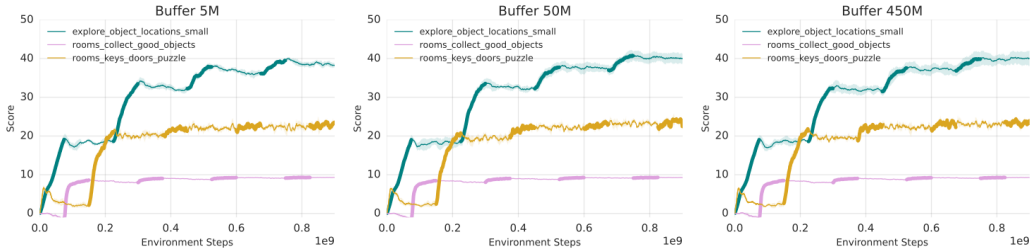

Figure 6: Comparison of the performance of training with different buffer sizes, using reservoir sampling to ensure that each buffer stores a uniform sample from all past experience. We observe only minimal difference in performance, with the smallest buffer (storing 1 in 200 experiences) demonstrating some catastrophic forgetting. See Appendix B for plots of the same data, showing cumulative performance.

We consider a sequence of tasks with 900 million environmental frames, comparing a large buffer of capacity 450 million to two small buffers of capacity 5 and 50 million. We find that all buffers perform well and conclude that it is possible to learn and reduce catastrophic forgetting even with a replay buffer that is significantly smaller than the total number of experiences. Decreasing the buffer size to 5 million results in a slight decrease in robustness to catastrophic forgetting. This may be due to over-fitting to the limited examples present in the buffer, on which the learner trains disproportionately often.

## 4.6 Comparison to P&C and EWC

Finally, we compare our method to Progress & Compress (P&C) [29] and Elastic Weight Consolidation (EWC) [13], state-of-the-art methods for reducing catastrophic forgetting that, unlike replay,

assume that the boundaries between different tasks are known (Figure 7). We use exactly the same sequence of Atari tasks as the authors of P&C [29], with the same time spent on each task. Likewise, the network and hyperparameters we use are designed to match exactly those used in [29]. This is simplified by the authors of P&C also using a training paradigm based on that in [3]. In this case, we use CLEAR with a 75-25 balance of new-replay experience.

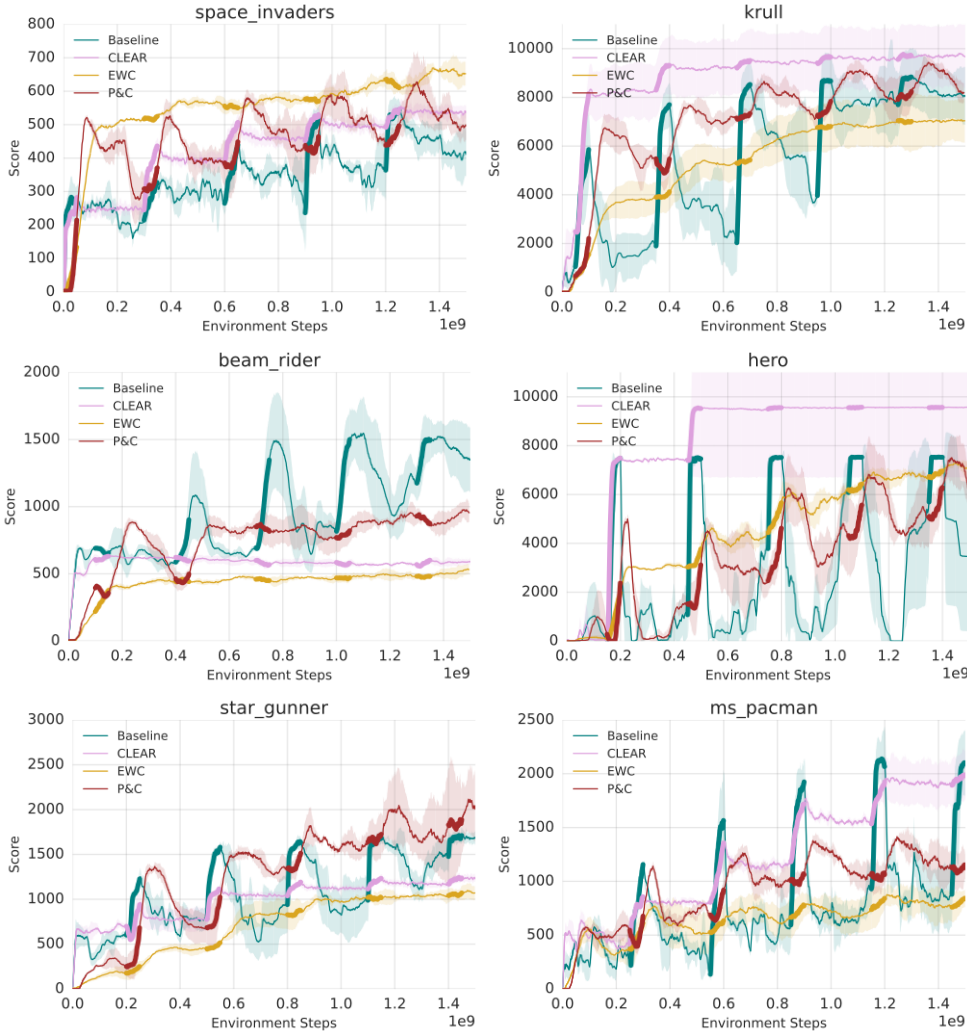

| | space_invaders | krull | beam_rider | hero | star_gunner | ms_pacman |
|---|---|---|---|---|---|---|
| Baseline | 346.47 | 5512.36 | **952.72** | 2737.32 | 1065.20 | 753.83 |
| CLEAR | 426.72 | **8845.12** | 585.05 | **8106.22** | 991.74 | **1222.82** |
| EWC | **549.22** | 5352.92 | 432.78 | 4499.35 | 704.20 | 639.75 |
| P&C | 455.34 | 7025.40 | 743.38 | 3433.10 | **1261.86** | 924.52 |

Figure 7: (a) Comparison of CLEAR to Progress & Compress (P&C) and Elastic Weight Consolidation (EWC). We find that CLEAR demonstrates comparable or greater performance than these methods, despite being significantly simpler and not requiring any knowledge of boundaries between tasks. See Appendix B for plots of the same data, showing cumulative performance. (b) Quantitative comparison of the final cumulative performance between these methods. Overall, CLEAR performs comparably to or better than P&C, and significantly better than EWC and baseline.

We find that we obtain slightly better performance than P&C and significantly better performance than EWC, despite CLEAR being much simpler and agnostic to the boundaries between tasks. It is worth

noting that though the on-policy model (baseline) experiences significant catastrophic forgetting, it also rapidly re-acquires its previous performance after re-exposure to the task; this allows baseline to be cumulatively better than EWC on some tasks (as is noted in the original paper [13]). An alternative plot of this experiment, showing cumulative performance on each task, is presented in Appendix B.

## 5   Discussion

The ingredients in the CLEAR method are intuitively complementary. The behavioral cloning objective keeps the policy distribution close to historical data, which in turn ensures that the importance-weighted gradient estimates have lower variance than they would with purely off-policy estimation. The behavioral cloning loss acts as regularization while not preventing ultimate convergence; as training proceeds, the behavioral cloning loss increasingly acts to regularize the network towards improved policies, which constitute a greater proportion of the replayed experience. Further study is warranted to study these intuitive claims more formally.

Algorithms for continual learning may live on a *Pareto frontier*: different methods may have different regimes of applicability. In cases where storing a memory buffer is truly prohibitive, methods that protect inferred parameters, such as Progress & Compress, may be more suitable. When task identities are available or boundaries between tasks are clear, leveraging this information may reduce memory or computational demands or be useful to alert the agent to learn rapidly. Further, there exist training scenarios that are adversarial either to our method or to any method that prevents forgetting. For example, if the action space of a task changed during training, fitting to the old policy's action distribution, through behavioral cloning, off-policy learning, weight protection, or any of a number of other strategies for preventing forgetting, could have a deleterious effect on performance. For such cases, we may need to develop algorithms that selectively *forget* skills as well as protecting them.

We anticipate many algorithmic innovations that build on the ideas set forward here. For example, weight-consolidation techniques such as Progress & Compress are quite orthogonal to our approach and could be married with it for further performance gains. Moreover, while the V-Trace algorithm we use is effective at off-policy correction for small shifts between the present and past policy distributions, it is possible that off-policy approaches leveraging Q-functions, such as Retrace [21], may prove more powerful still. We believe the experimental protocols we introduce – such as comparing separate, simultaneous, and sequential training, and "probing" with a novel task at different points in a training sequence – may prove useful in guiding future work in this area.

We have described a simple but powerful approach for preventing catastrophic forgetting in continual learning settings. CLEAR uses on-policy learning on fresh experiences to adapt rapidly to new tasks, while using off-policy learning with behavioral cloning on replay experience to maintain and modestly enhance performance on past tasks. Behavioral cloning on replay data further enhances the agent's stability. Our method is simple, scalable, and practical; it takes advantage of the general abundance of memory and storage in modern computers and computing facilities. We believe that the broad applicability and simplicity of the approach make CLEAR a candidate "first line of defense" against catastrophic forgetting in many RL contexts.

## Footnotes

[1]As an example of (destructive) interference, learning how to drive on the right side of the road may be difficult while also learning how to drive on the left side of the road, because of the nature of the tasks involved. Forgetting, by contrast, results from sequentially training on one and then another task - e.g. learning how to drive and then not doing it for a long time.

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
