[Supplementary Material]

# A  Implementation details

## A.1  Distributed setup

Our training setup was based on that of [3], with multiple actors and a single learner. The actors (which run on CPU) generate training examples, which are then sent to the learner. Weight updates made by the learner are propagated asynchronously to the actors. The workflows for each actor and for the learner are described below in more detail.

**Actor.** A training episode (*unroll*) is generated and inserted into the actor's buffer. Reservoir sampling is used (see details below) if the buffer has reached its maximum capacity. The actor then samples another unroll from the buffer. The new unroll and replay unroll are both fed into a queue of examples that are read by the learner. The actor waits until its last example in the queue is read before creating another.

**Learner.** Each element of a batch is a pair (new unroll, replay unroll) from the queue provided by actors. Thus, the number of new unrolls and the number of replay unrolls both equal the entire batch size. Depending on the buffer utilization hyperparameter (see Figure 5), the learner uses a balance of new and replay examples, taking either the new unroll or the replay unroll from each pair. Thus, no actor contributes more than a single example to the batch (reducing the variance of batches).

## A.2  Network

For our DMLab experiments, we used the same network as in the DMLab experiments of [3]. We selected the shallower of the models considered there (a network based on [20]), omitting the additional LSTM module used for processing textual input since none of the tasks we considered included such input. For Atari, we used the same network in Progress & Compress [29] (which is also based on [3]), also copying all hyperparameters.

## A.3  Buffers

Our replay buffer stores all information necessary for the V-Trace algorithm, namely the input presented by the environment, the output logits of the network, the value function output by the network, the action taken, and the reward obtained. Leveraging the distributed setup, the buffer is split among all actors equally, so that, for example, if the total buffer size were one million across a hundred actors, then each actor would have buffer capacity of ten thousand. All buffer sizes are measured in environment frames (not in numbers of unrolls), in keeping with the $x$-axis of our training plots. For baseline experiments, no buffer was used, while all other parameters and the network remained constant.

Unless otherwise specified, the replay buffer was capped at half the number of environment frames on which the network is trained. This is by design – to show that even past the buffer capacity, replay continues to prevent catastrophic forgetting. When the buffer fills up, then new unrolls are added by reservoir sampling, so that the buffer at any given point contains a uniformly random sample of all unrolls up until the present time. Reservoir sampling is implemented as in [10] by having each unroll associated with a random number between 0 and 1. A threshold is initialized to 0 and rises with time so that the number of unrolls above the threshold is fixed at the capacity of the buffer. Each unroll is either stored or abandoned in its entirety; no unroll is partially stored, as this would preclude training.

## A.4  Training

Training was conducted using V-Trace, with hyperparameters on DMLab/Atari tasks set as in [3]. Behavioral cloning loss functions $L_{\text{policy-cloning}}$ and $L_{\text{value-cloning}}$ were added in some experiments with weights of 0.01 and 0.005, respectively. The established loss functions $L_{\text{policy-gradient}}$, $L_{\text{value}}$, and $L_{\text{entropy}}$ were applied with weights of 1, 0.5, and $\approx 0.005$, in keeping with [3]. No significant effort was made to optimize fully the hyperparameters for CLEAR.

## A.5  Evaluation

We evaluate each network during training on all tasks, not simply that task on which it is currently being trained. Evaluation is performed by pools of testing actors, with a separate pool for each task

in question. Each pool of testing actors asynchronously updates its weights to match those of the learner, similarly to the standard (training) actors used in our distributed learning setup. The key differences are that each testing actor (i) has no replay buffer, (ii) does not feed examples to the learner for training, (iii) runs on its designated task regardless of whether this task is the one currently in use by training actors.

## A.6 Experiments

In many of our experiments, we consider tasks that change after a specified number of learning episodes. The total number of episodes is monitored by the learner, and all actors switch between tasks simultaneously at the designated point, henceforward feeding examples to the learner based on experiences on the new task (as well as replay examples). Each experiment was run independently three times; figures plot the mean performance across runs, with error bars showing the standard deviation.

## B Figures replotted according to cumulative sum

In this section, we replot the results of our main experiments, so that the $y$-axis shows the mean cumulative reward obtained on each task during training; that is, the reward shown for time $t$ is the average $(1/t)\sum_{s<t} r_s$. This makes it easier to compare performance between models, though it smoothes out the individual periods of catastrophic forgetting so they are no longer visible. Figures 3 and 7 in the main body of the paper tabulate the final cumulative rewards at the end of training.

Figure 8: Alternative plot of the experiments shown in Figure 1, showing the difference in cumulative performance between training on tasks separately, simultaneously, and sequentially (without using CLEAR). The marked decrease in performance for sequential training is due to catastrophic forgetting. As in our earlier plots, thicker line segments are used to denote times at which the network is gaining new experience on a given task.

Figure 9: Alternative plot of the experiments shown in Figure 2, showing how applying CLEAR when training on sequentially presented tasks gives almost the same results as training on all tasks simultaneously (compare to sequential and simultaneous training in Figure 8 above). Applying CLEAR without behavioral cloning also yields decent results.

Figure 10: Alternative plot of the experiments shown in Figure 5, comparing performance between using CLEAR with 75-25 new-replay experience, 50-50 new-replay experience, and 100% replay experience. An equal balance of new and replay experience seems to represent a good tradeoff between stability and plasticity, while 100% replay reduces forgetting but lowers performance overall.

Figure 11: Alternative plot of the experiments shown in Figure 6, showing that reduced-size buffers still allow CLEAR to achieve essentially the same performance.

Figure 12: Alternative plot of the experiments shown in Figure 7, showing that CLEAR attains comparable or better performance than the more complicated methods Progress & Compress (P&C) and Elastic Weight Consolidation (EWC), which also require information about task boundaries, unlike CLEAR.