[Reviews · NeurIPS 2019]

Reviewer 1



Originality: The proposed algorithm and analyses within the submission are variants of work done in the area in a new combination. Thus, the work is not groundbreaking in terms of its novelty. Quality: Although the novelty of the submission isn't groundbreaking, the submission makes up for it in the quality of the analyses. The authors put together interesting current ideas together in a manner that is new and seems to work really nicely. The set of analyses illustrate the strength and limitations of different components of their method and how it compares to previous work. My only concern is how sensitive the method is to the number of learned tasks. Why did you pick those three? How is your method affected by the number of tasks it needs to learn? Is there a tradeoff in the number of examples needed per task as the number of tasks grows? A minor comment: I was not convinced that the authors had disentangled interference and catastrophic forgetting (Lines 109-122). How is changing weights due to "task incompatibility" vs."overwriting" occurring due to learning a new task different? The distinction referenced by the multi-task structure seems to me to more mirror batch vs. interleaved learning from the cognitive science literature than interference and forgetting. Clarity: The paper is well-written, albeit a bit challenging to access as an outsider to multi-task RL. If possible, it would be helpful to briefly describe some of the multi-task RL jargon that isn't standard across the rest of the NeurIPS literature (e.g., what is meant by "rollout" and "Importance Weighted Actor-Learner" in this context). I say if possible, because I'm aware of the space constraints. Significance: I think this could be a highly important paper for the field, and inspire a number of extensions by different researchers. Response to author feedback: I appreciate the thoughtful responses from the authors with regards to my initial review. I am still have some concern with regards to the arbitrariness of some choices (especially after having a chance to see the other reviews as well), but I am satisfied enough that I vote for its acceptance. With respect to the forgetting and interference, I hope you clarify these two terms a bit more than in your feedback. The issue is that one of the major theories for forgetting within the psychology memory literature is due to interference. I appreciate that the authors will be revising the paper to ensure it is properly understood across disciplines.

Reviewer 2



Quality: Given the impressive performance of the method in reducing catastrophic forgetting, even without labeling of individual tasks, this seems to be a significant result. In addition, the authors are thorough with their experiments, considering several parameters, and present a reasonable level of understanding of how the method is working. Clarity: While I am not an expert in the field, I found the description of the method, in particular the description of the V-trace target, to be unnecessarily dense. I would have very much benefitted from some more detail and intuition behind the first equation, especially the truncation of importance sampling weights. I found the behavioral cloning elements very clear, and the other components would benefit from similar exposition. I found the discussion of the experiments to be very clear. Originality and Significance: I think the advances in this paper are both original and significant.

Reviewer 3



The authors propose CLEAR, a simple approach to continual reinforcement learning: storing old trajectories. They use a state-of-the-art algorithm for control, IMPALA (which comes with its own off-policy-ness correction method: V-trace), to combine learning from current experiences with off-policy learning from old trajectories stored in memory. The authors add two extra costs for behavioral cloning: a KL divergence between stored and online policies, and the squared distance between stored and current state values that also help alleviate catastrophic forgetting. The article evaluates the proposed method on two setups demonstrating both stability (no forgetting) and plasticity (no resistance to learning new tasks). CLEAR achieves state-of-the-art performance on a sequence of Atari games. Since the empirical evaluation would represent the significant contribution of this article, I consider that the experiments presented still leave some unanswered questions regarding how much data is needed, if the choice of algorithm is important (besides having a way of learning off-policy), or if the number of interactions with each task is important. Therefore I suggest a weak reject decision. The strong aspects of this paper are: the eloquent exposure of the proposed method, and the extensive set of experiments meant to clear aspects regarding task interference vs. forgetting, the ratio of old / new experiences, the memory size, and comparison with other algorithms. Since the proposed idea is not new, and the conclusions are not surprising in any way, the empirical validation of the different claims is the most important in the article. I think that in some aspects the set of experiments is not entirely satisfactory and the article could benefit from answering some additional questions: - Most of the experiments are conducted only on a sequence of three cyclically repeating tasks. It might be interesting to see how it works for a larger number of tasks, or for tasks that are not being revisited several times. Also, the agent is trained on the same number of experiences on all tasks. This strategy fills the memory with a convenient distribution of trajectories and makes it easier to claim that CLEAR needs no information on task boundaries. What happens if the agent spends disproportionate amounts of time in the tasks? - There are several algorithms capable of learning from off-policy data. The current study would be stronger if it evaluates additional algorithms beside IMPALA in this continual learning setup, such as Retrace or Soft Actor-Critic. - Also, it’s nice that the authors disentangled task interference and forgetting, but the proposed sequence of tasks did not exhibit destructive interference. It would be useful to know what happens when this phenomenon occurs and learning must lean either towards stability or plasticity. - There are a couple of experiments with memories of different sizes. Not surprisingly, reducing the memory size too much makes the algorithm inefficient in avoiding forgetting old tasks. It would be informative to explore the dependence between the performance on old tasks and the memory size (is it linear, exponential, …?). - why do authors choose the 75-25 ratio for comparison with EWC and P&C since this was not the best configuration in the experiments in DMlab Originality The idea of keeping data from previous tasks is not that original, but neither the authors claim that. The effort was put into evaluating the method. Quality The article proposes a method, details a few aspects that are worth to be explored, designs appropriate experiments and presents the results in an eloquent manner. Clarity The article is very well written with an easy to follow structure. Significance Although the idea is simple, such a study would be of great importance for the continual learning community. If it would fully answer the question “when and in what limits of memory is keeping data a sufficient solution for continual learning?” then the article would set an important baseline for more sophisticated methods. But my opinion is that it lacks some quantitative experiments to fully answer the aforementioned question.

Reviewer 4



The issue that I found with the paper is regarding the initial formalization of the approach. I mean the technical details. It is really poor. Indeed it takes only three quaters of Page 3. I have also some concerns about the exposition of the paper. The terms like "behavioural cloning", "V-Trace", "historical policy distribution" and so on. I am also confused by the lack of interpretation of the empirical results. One can indeed all kinds of combinations and plot the results. But it does not add any value unless one can technically interpret the behaviour. That aspect is lacking in the paper. One needs to come up with some basic justification for the kind of plasticity or stability or robustness achieved in the results.

Reviewer 5



On the positive side, the paper is well-written, the ideas and algorithm are fairly straightforward, and the performance of the algorithm is good. I also thought that the discussion and accompanying experiments distinguishing between interference and forgetting was very nice, and as the authors point out these experimental protocols can be useful for future studies of continual learning. On the negative side, I felt that this advance was somewhat incremental, especially given that the advantage of this method over existing ones was not clearly demonstrated (see below). I should say that I'm not a stickler for state-of-the-art if the paper's ideas are highly original, but I don't think that is the case here; the main selling point of the algorithm is its simplicity. On this point, though, I wasn't sure whether it's really that important that the algorithm doesn't need task boundaries. It seems to me that these boundaries are available for most tasks. In any case, it was interesting the algorithm could perform well without knowing the boundaries. Detailed comments: Is 50-50 new-replay always the best tradeoff? Can this percentage be optimized by the agent? The cumulative performance table shown in Fig 7 is all over the place, with different tasks favoring different algorithms. I don't see any clear pattern here favoring CLEAR. Moreover, on those tasks where it does come out on top, the error bars are overlapping with those of other algorithms (though this is hard to interpret because the caption doesn't say what the error bars are showing; are these standard errors?). Minor comments: p. 7: "that P&C" -> "than P&C" Text in figures is not legible.

[Author Response · NeurIPS 2019]

We are grateful to the reviewers for their careful reading and thoughtful comments. Reviewer #1 states that "The two contributions are from in-depth analyses and thus, are substantially more rich than the standard submission...I think this could be a highly important paper for the field, and inspire a number of extensions by different researchers." Reviewer #2 writes "I found this to be a good paper and enjoyed reading it" and "I found the discussion of the experiments to be very clear." Reviewer #3 notes that the paper "although the idea is simple, such a study would be of great importance for the continual learning community" while Reviewer #5 writes that we "show empirically on some strong experiments that [our algorithm] has both stability and plasticity."

We agree with the reviewers that our paper offers (i) a simple method for preventing catastrophic forgetting in continual learning that makes no assumptions about task boundaries (unlike other algorithms), (ii) innovative empirical analyses that set new standards for probing the properties of continual learning algorithms, (iii) demonstrations on hard problems, including standard benchmarks in this area, that we achieve both stability and plasticity, surpassing state-of-the-art. We respond to particular comments and questions below.

**Reviewer #1.** *(a) Task setup.* "Why did you pick those three?" They represent very distinct tasks within the DMLab suite. "How is your method affected by the number of tasks it needs to learn?" As shown in Figures 4 and 7, it is possible for the method to perform well up to (at least) six tasks, which was the maximum we tested with. "Is there a tradeoff in the number of examples needed per task as the number of tasks grows?" Figure 4 (plasticity) shows that the time required to learn a new task does not immediately degrade as the number of tasks grows, though clearly this will not hold for arbitrarily many tasks. Please also see our response to Rev. #3, point (a), below.

*(b) Interference and forgetting.* We will make this difference clearer in the text: we see interference as a consequence of the nature of the tasks involved making them more difficult to learn together than separately - e.g. learning how to drive on the right side of the road may be difficult while also learning how to drive on the left side of the road. Forgetting, by contrast, results from sequentially training on one and then another task - e.g. learning how to drive and then not doing it for a long time. Most pairs of tasks seem to interfere minimally, if at all, while forgetting occurs in deep RL for essentially any pair of tasks.

*(c) RL terminology.* We will make this clearer in the revision, expanding on the definitions of terms such as "rollout" and "Importance Weighted Actor-Learner" and giving intuition for the V-Trace algorithm.

**Reviewer #2.** We will make the exposition of RL techniques much clearer in the revision, as noted above. Thank you for calling this to our attention. In particular, we will provide greater description and intuition for the V-Trace algorithm, in addition to being more explicit about overall RL terminology.

**Reviewer #3.** *(a) Task setup.* The cyclic training paradigm is versatile, and we wanted to explore our method thoroughly in the simplest non-trivial forgetting scenario. Note that the agent does indeed spend a very large amount of time (75 million environment steps) on each task before switching to the next, causing, as we demonstrate, baseline RL algorithms to thoroughly forget each task. Please also see our response to Rev. #1, point (a). Other task setups would indeed be interesting to consider in future work, but we also think this paradigm merits adoption in subsequent papers.

*(b) Other off-policy algorithms.* Yes, we could use a different off-policy RL algorithm – though the demonstration of the principle and its utility would be largely the same. Rather than compare multiple off-policy algorithms, we focused on exploring performance in different experimental setups. Making Retrace work is an exciting avenue for more work in the future. We've now commented in the paper how our algorithm can be straightforwardly combined with stronger off-policy learning algorithms.

*(c) Destructive interference.* We were trying to fix the specific problem of forgetting, and therefore tried to disentangle these various effects by minimizing both constructive and destructive interference. While very minimal constructive interference resulted naturally from tasks being drawn from the same suite, we expect that destructive interference is a less common phenomenon that one may explicitly need to seek out, arising primarily when two tasks impose conflicting constraints on the learner. See Figure 1 and Subsection 4.1.

*(d) Small memory buffers.* Surprisingly, we actually didn't observe any significant decrease in performance from reducing the memory size (see Figure 6), even with a buffer as little as 0.5% of past experience. Exploring what happens at even smaller buffers would indeed be an interesting future direction.

*(e) New-replay ratio.* For Atari, 75-25 performed slightly better than 50-50, but in both DMLab and Atari, both 75-25 and 50-50 worked very well (see Figure 5).

**Reviewer #5.** We would respectfully disagree that the paper is missing an interpretation of empirical results or justification for the plasticity and stability properties we observe. We would be very happy to address specific criticisms or failures in clarity in the revision. We will certainly provide more background for terms from the RL literature such as "behavioural cloning", "V-Trace", and "historical policy distribution" (see Rev. #1, point (c)).

[Meta-Review · NeurIPS 2019]

The paper addresses the problem of catastrophic forgetting when learning different tasks in RL. The proposed approach is based on experience replay. While the approach is of moderate novelty, it has the interesting property compared to the more complex approaches at the state of the art (e.g. Progress and Compress and Elastic Weight Consolidation) that it does not require the tasks and their boundaries to be known beforehand. The bulk of the paper is on the experiments, considering three DMLab tasks. It is shown that the efficiency of the approach does not depend on the fraction of past/novel tasks, and on the size of the memory buffer. Complementary investigations are required in the camera-ready to better understand how and why it works: * using a visualization of the internal state of the network, to understand whether the learner stores the policy for the different tasks in different regions of the (latent) state space; * assessing the sample complexity of each task and setting the size of the memory buffer to 1, 1/2, 1/3 of this sample complexity; * examining the sensitivity of the approach to the truncated importance sampling coefficients \bar c and \bar \rho.